# Celebrating Diversity in Shared Multi-Agent Reinforcement Learning

**Chenghao Li, Tonghan Wang, Chengjie Wu, Qianchuan Zhao, Jun Yang\*, Chongjie Zhang\***
Tsinghua University
{lich18, wangth18, wucj19}@mails.tsinghua.edu.cn,
{zhaoqc, yangjun603, chongjie}@tsinghua.edu.cn

## Abstract

Recently, deep multi-agent reinforcement learning (MARL) has shown the promise to solve complex cooperative tasks. Its success is partly because of parameter sharing among agents. However, such sharing may lead agents to behave similarly and limit their coordination capacity. In this paper, we aim to introduce diversity in both optimization and representation of shared multi-agent reinforcement learning. Specifically, we propose an information-theoretical regularization to maximize the mutual information between agents' identities and their trajectories, encouraging extensive exploration and diverse individualized behaviors. In representation, we incorporate agent-specific modules in the shared neural network architecture, which are regularized by L1-norm to promote learning sharing among agents while keeping necessary diversity. Empirical results show that our method achieves state-of-the-art performance on Google Research Football and super hard StarCraft II micromanagement tasks[†].

## 1 Introduction

Cooperative multi-agent reinforcement learning (MARL) has drawn increasing interest in recent years, which provides a promise for solving many real-world challenging problems, such as sensor networks [1], traffic management [2], and coordination of robot swarms [3]. However, learning effective policies for such complex multi-agent systems remains challenging. One central problem is that the joint action-observation space grows exponentially with the number of agents, which imposes high demand on the scalability of learning algorithms.

To address this scalability challenge, *policy decentralization with shared parameters* (PDSP) is widely used, where agents share their neural network weights. Parameter sharing significantly improves learning efficiency because it dramatically reduces the total number of policy parameters, while experiences and gradients of one agent can be used to train others. Enjoying these advantages, many advanced deep MARL approaches adopt the PDSP paradigm, including value-based methods [4–8], policy gradients [9–13] and communication learning algorithms [14, 15]. These approaches achieve state-of-the-art performance on tasks such as StarCraft II micromanagement [16].

While parameter sharing has been proven to accelerate training [17], its drawbacks are also apparent in complex tasks. These tasks typically require substantial exploration and diversified strategies among agents. When parameters are shared, agents tend to acquire homogeneous behaviors because they typically adopt similar actions under similar observations, preventing efficient exploration and the emergence of sophisticated cooperative policies. This tendency becomes particularly problematic

---

\*Equal advising

[†]Videos are available at `https://sites.google.com/view/celebrate-diversity-shared` with codes.

for many challenging multi-agent coordination tasks, hindering deep MARL from broader applications. For example, the unsatisfactory performance of state-of-the-art MARL algorithms on Google Research Football (Fig. 1, and [18]) highlights an urgent demand for diverse behaviors.

Notably, sacrificing the merits of parameter sharing for diversity is also unfavorable. Like humans, sharing necessary experience or understanding of tasks can broadly accelerate cooperation learning. Without parameter sharing, agents search in a much larger parameter space, which may be wasteful because they do not need to behave differently all the time. Therefore, the question is how to adaptively trade-off diversity and sharing. In this paper, we solve this dilemma by proposing several structural and learning novelties.

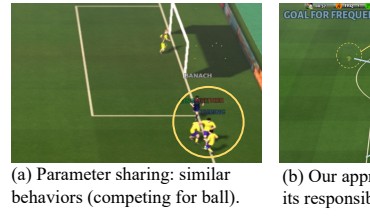
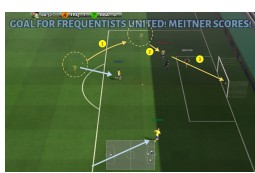

(a) Parameter sharing: similar behaviors (competing for ball).

(b) Our approach: each agent has its responsibility to score.

- Agents' history location
- The goal's passing, dribbling and shooting
- Off-the-ball moving

Figure 1: Shared parameters induce behaviors (**left**) and can hardly learn successful policies on the challenging Google Research Football task. Our method learns sophisticated cooperative strategies by **trading off diversity and sharing** (**right**).

To encourage diversity, we propose a novel information-theoretical objective to maximize the mutual information between agents' identities and trajectories. This objective enables each agent to distinguish themselves from others and thus involves the contribution of all agents. Accordingly, we derive an intrinsic reward for motivating diversity and optimize it with the global environmental reward by learning the total Q-function as a combination of individual Q-functions. Structurally, we further decompose individual Q-functions as the sum of shared and non-shared local Q-functions for sharing experiences while maintaining representation diversity. We hope agents can use and expand shared knowledge whenever possible. Thus we introduce L1 regularization on each non-shared Q-function, encouraging agents to share and be diverse when necessary on several critical actions. Combining these novelties achieves a dynamic balance between diversity and homogeneity, efficiently catalyzing adaptive and sophisticated cooperation.

We benchmark our approach on Google Research Football (GRF) [18], and StarCraft II micromanagement tasks (SMAC) [16]. The extraordinary performance of our approach on challenging benchmarking tasks shows that our approach achieve significantly higher coordination capacity than baselines while using diversity as a catalyst for more robust and talent policies. To our best knowledge, our approach achieves state-of-the-art performance on SMAC super hard maps and challenging GRF multi-agent tasks like `academy_3_vs_1_with_keeper`, `academy_counterattack_hard`, and a full-field scenario `3_vs_1_with_keeper` (full field).

## 2 Background

A fully cooperative multi-agent task can be formulated as a Dec-POMDP [19], which is defined as a tuple $\mathcal{G} = \langle N, S, A, P, R, O, \Omega, n, \gamma \rangle$, where $N$ is a finite set of $n$ agents, $s \in S$ is the true state of the environment, $A$ is the set of actions, and $\gamma \in [0, 1)$ is a discount factor. At each time step, each agent $i \in N$ receives his own observation $o_i \in \Omega$ according to the observation function $O(s, i)$, and selects an action $a_i \in A$, which results in a joint action vector $\boldsymbol{a}$. The environment then transitions to a new state $s'$ based on the transition function $P(s'|s, \boldsymbol{a})$, and inducing a global reward $r = R(s, \boldsymbol{a})$ shared by all the agents. Each agent has its own action-observation history $\tau_i \in \mathcal{T}_i \doteq (\Omega_i \times A)^*$. Due to partial observability, each agent conditions its policy $\pi_i(a_i|\tau_i)$ on $\tau_i$. The joint policy $\boldsymbol{\pi}$ induces the joint action-value function $Q_{tot}^{\boldsymbol{\pi}}(s, \boldsymbol{a}) = \mathbb{E}_{s_{0:\infty}, \boldsymbol{a}_{0:\infty}} [\sum_{t=0}^{\infty} \gamma^t r_t \mid s_0 = s, \boldsymbol{a}_0 = \boldsymbol{a}, \boldsymbol{\pi}]$.

### 2.1 Centralized Training with Decentralized Execution

Our method adopts the framework of centralized training with decentralized execution (CTDE) [9, 20, 4, 5, 21, 22, 6, 11]. This framework tackles the exponentially growing joint action space by decentralizing the control policies while adopting centralized training to learn cooperation. Agents learn in a centralized manner with access to global information but execute based on their local action-observation history. One promising approach to implement the CTDE framework is value function factorization. The IGM (individual-global-max) principle [21] guarantees the consistency between the local and global greedy actions. When IGM is satisfied, agents can obtain the optimal

global action by simply choosing the local greedy action that maximizes each agent's individual utility function $Q_i$. Some algorithms have successfully used the IGM principle [5, 6, 23] to push forward the progress of MARL.

## 3 Method

In this section, we present a novel diversity-driven MARL framework (Fig. 2) that balances each agent's individuality with group coordination, which is a general approach that can be combined with existing CDTE value factorization methods.

### 3.1 Identity-Aware Diversity

We first introduce how to encourage behavioral diversity by designing intrinsic motivations. Intuitively, to encourage the specialty of individual trajectories, agents need to behave differently to highlight themselves from others, taking different actions and visiting different local observations. To achieve this goal, we use an information-theoretic objective for maximizing the mutual information between individual trajectory and agents' identity:

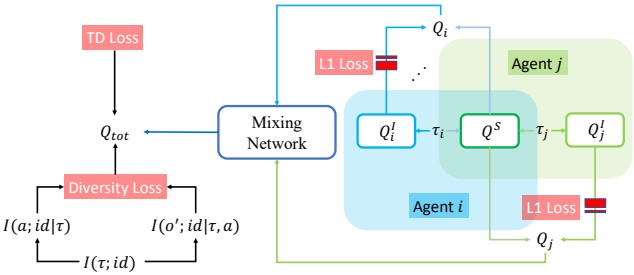

Figure 2: Schematics of our approach.

$$I^{\boldsymbol{\pi}}(\tau_T; id) = H(\tau_T) - H(\tau_T|id) = E_{id,\tau_T \sim \boldsymbol{\pi}}\left[\log \frac{p(\tau_T|id)}{p(\tau_T)}\right], \tag{1}$$

where $\tau_T$ and $id$ is the random variable for agent's local trajectory and identity, respectively. $\boldsymbol{\pi}$ is the joint policy. To optimize Eq. 1, we expand $p(\tau_T)$ as $p(o_0)\prod_{t=0}^{T-1} p(a_t|\tau_t)p(o_{t+1}|\tau_t, a_t)$, and $p(\tau_T|id)$ as $p(o_0|id)\prod_{t=0}^{T-1} p(a_t|\tau_t, id)p(o_{t+1}|\tau_t, a_t, id)$. Therefore, the mutual information can be written as:

$$I^{\boldsymbol{\pi}}(\tau_T; id) = E_{id,\tau}\left[\underbrace{\log \frac{p(o_0|id)}{p(o_0)}}_{①} + \underbrace{\sum_{t=0}^{T-1} \log \frac{p(a_t|\tau_t, id)}{p(a_t|\tau_t)}}_{②} + \underbrace{\sum_{t=0}^{T-1} \log \frac{p(o_{t+1}|\tau_t, a_t, id)}{p(o_{t+1}|\tau_t, a_t)}}_{③}\right]. \tag{2}$$

Term ① is determined by the environment, and we can ignore it when optimizing the mutual information. The second term quantifies the information gain about agent's action selection when the identity is given, which measures **action-aware diversity** as $I(a; id|\tau)$. However, $p(a_t|\tau_t, id)$ is typically the distribution induced by $\epsilon$-greedy, which only distinguishes the action with the highest possibility. Therefore, directly optimizing this term conceals most information about the local Q-functions. To solve this problem, we use the Boltzmann softmax distribution of local Q values to replace $p(a_t|\tau_t, id)$, which forms a lower bound of term ②:

$$E_{id,\tau}\left[\log \frac{p(a_t|\tau_t, id)}{p(a_t|\tau_t)}\right] \geq E_{id,\tau}\left[\log \frac{\text{SoftMax}(\frac{1}{\alpha}Q(a_t|\tau_t, id))}{p(a_t|\tau_t)}\right]. \tag{3}$$

The inequity holds because the KL divergence $D_{\text{KL}}(p(\cdot|\tau_t, id)\|\text{SoftMax}(\frac{1}{\alpha}Q(\cdot|\tau_t, id)))$ is non-negative. We maximize this lower bound to optimize Term ②. Inspired by variational inference approaches [24], we derive and optimize a tractable lower bound for Term ③ at each timestep by introducing a variational posterior estimator $q_\phi$ parameterized by $\phi$:

$$E_{id,\tau}\left[\log \frac{p(o_{t+1}|\tau_t, a_t, id)}{p(o_{t+1}|\tau_t, a_t)}\right] \geq E_{id,\tau}\left[\log \frac{q_\phi(o_{t+1}|\tau_t, a_t, id)}{p(o_{t+1}|\tau_t, a_t)}\right], \tag{4}$$

Similar to the second term, the inequality holds because for any $q_\phi$, the KL divergence $D_{\text{KL}}(p(\cdot|\tau_t, a_t, id)\|q_\phi(\cdot|\tau_t, a_t, id))$ is non-negative. Intuitively, optimizing Eq. 4 encourages agents

to have diverse observations that are distinguishable by agents' identification and thus measures **observation-aware diversity** as $I(o'; id|\tau, a)$. To tighten the this lower bound, we minimize the KL divergence with respect to the parameters $\phi$. The gradient for updating $\phi$ is:

$$\nabla_\phi \mathcal{L}(\phi) = \nabla_\phi \mathbb{E}_{\tau,a,id} \left[ D_{\mathrm{KL}} \left( p\left(\cdot|\tau, a, id\right) \| q_\phi\left(\cdot|\tau, a, id\right) \right) \right] = \nabla_\phi \mathbb{E}_{\tau,a,id,o'} \left[ \log \frac{p\left(o'|\tau, a, id\right)}{q_\phi\left(o'|\tau, a, id\right)} \right]$$
$$= -\mathbb{E}_{\tau,a,id,o'} \left[ \nabla_\phi \log q_\phi\left(o'|\tau, a, id\right) \right]. \tag{5}$$

Based on the lower bounds shown in Eq. 3 and Eq. 4, we introduce intrinsic rewards to optimise the information-theoretic objective (Eq. 1) for encouraging diverse behaviors:

$$r^I = E_{id} \left[ \beta_2 D_{\mathrm{KL}}(\mathrm{SoftMax}(\beta_1 Q(\cdot|\tau_t, id)) \| p(\cdot|\tau_t)) \right. \\ \left. + \beta_1 \log q_\phi(o_{t+1}|\tau_t, a_t, id) - \log p(o_{t+1}|\tau_t, a_t) \right]. \tag{6}$$

We introduce two scaling factors $\beta_1, \beta_2 \geq 0$ when calculating intrinsic rewards. When $\beta_1$ is 0, we only optimize the entropy term $H(\tau_T)$ in the mutual information objective (Eq. 1). $\beta_2$ is used to adjust the importance of policy diversity compared with transition diversity. In Appendix A, we discuss and compare two different approaches for estimating $p\left(a_t|\tau_t\right)$ and $p\left(o_{t+1}|\tau_t, a_t\right)$.

### 3.2 Action-Value Learning for Balancing Diversity and Sharing

In the previous section, we introduce an information-theoretic objective for encouraging each agent to behave differently from general trajectories. However, the shared local Q-function does not have enough capacity to present different policies for each agent. For solving this problem, we additionally equip each agent $i$ with an individual local Q-function $Q_i^I$. Defining experiences that need to be shared or exclusively learned is inefficient and usually can not generalize. Therefore, we let agents adaptively decide whether to share experiences by decomposing $Q_i$ as:

$$Q_i(a_i|\tau_i) = Q^S(a_i|\tau_i) + Q_i^I(a_i|\tau_i), \tag{7}$$

where $Q^S$ is the shared Q-function among agents. In its current form, agents may learn to decompose their local Q-function arbitrarily. On the contrary, we expect that agents can share as much knowledge as possible so that we apply an L1 regularization on individual local Q-function $Q^I$ as shown in Fig.2. Such a regularization can also prevent agents from being too diverse and ignore cooperating to finish the task. In our experiments, we show that the L1 regularization is critical to achieving a balance between diversity and cooperation.

### 3.3 Overall Learning Objective

In this section, we discuss how to use the diversity-encouraging reward to train the proposed learning framework. Since the intrinsic rewards $r^I$ inevitably involves the influence from all agents, we add $r^I$ to environment rewards $r^e$ and use the following TD loss:

$$\mathcal{L}_{TD}(\theta) = \left[ r^e + \beta r^I + \gamma \max_{\boldsymbol{a}'} Q_{tot}\left(s', \boldsymbol{a}'; \theta^-\right) - Q_{tot}(s, \boldsymbol{a}; \theta) \right]^2, \tag{8}$$

where $\theta$ is the parameters in the whole framework, $\theta^-$ is periodically frozen parameters copied from $\theta$ for a stable update, and $\beta$ is a hyper-parameter adjusting the weight of intrinsic rewards compared with environment rewards. We use QPLEX to decompose $Q_{tot}$ as mixing of local Q-functions $Q_i$ and train the framework end-to-end by minimizing the loss:

$$\mathcal{L}(\theta) = \mathcal{L}_{TD}(\theta) + \lambda \sum_i \mathcal{L}_{L_1}(Q_i^I(\theta_i^I)), \tag{9}$$

where $\theta_i^I$ is the parameters of $Q_i^I$, $\mathcal{L}_{L_1}(Q_i^I)$ is the L1 regularization term for independent Q-functions, and $\lambda$ is a scaling factor.

## 4 Case study: outperforming by being diverse only when necessary

We design `Pac-Men` shown in Fig. 3 to demonstrate how our approach works. In this task, four agents are initialized at the center room and can only observe a $5 \times 5$ grid around them. Three dots

are initialized randomly in each edge room. To make this environment more challenging, paths to different rooms have different lengths, which are down : left : up : right = 4 : 8 : 12 : 8. Three out of four paths are outside agents' observation scope, which brings about the difficulty of exploration. Dots will refresh randomly after all rooms are empty. An ineffective competition between agents occurs when they come together in one room. The total environmental reward is the number of dots eaten in one step or -0.1 if no one eats dots. The time limit of this environment is set to 100 steps.

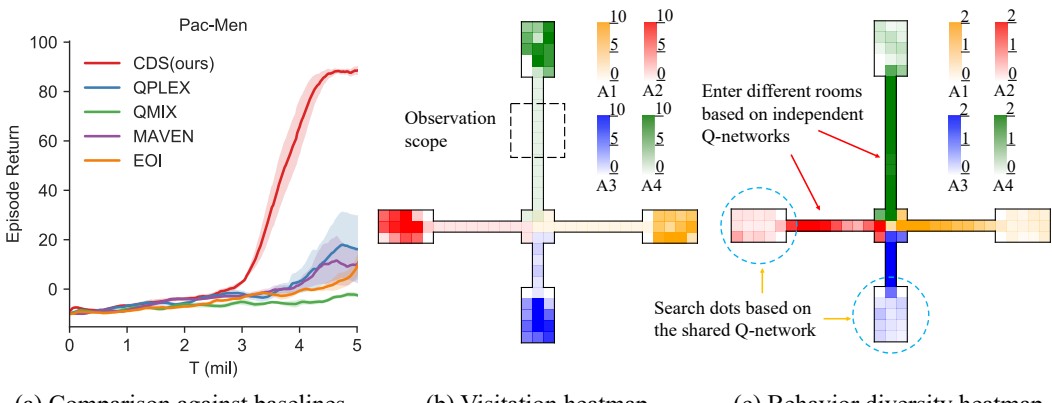

(a) Comparison against baselines      (b) Visitation heatmap      (c) Behavior diversity heatmap

Figure 3: Why does our method work? The balance between identity-aware diversity and experience sharing encourages sophisticated strategies.

Fig. 3-middle demonstrates the learned strategies of our approach, with a heatmap showing the visitation number. Driven by the objective of mutual information between individual trajectory and identity, agents achieve diversity and scatter in different rooms to eat dots. We further analyze the role of independent and shared Q-functions during different stages in Fig. 3 right. We visualize the value of $SD(Q_i^I(\cdot))/SD(Q^S(\cdot))$, where SD denotes the standard deviation (SD) of Q values for different actions. A higher SD ratio indicates the independent Q-functions play a leading role, while a lower SD ratio indicates the shared Q function's domination.

We notice that the SD ratio is considerably larger in the central room and four paths than in four edge rooms. This observation means that agents use independent Q networks to reach different rooms while use the shared Q network to search for dots in them. The result shows that our method achieves a good balance between diversity and knowledge sharing. Taking this advantage, our approach outperforms baselines (Fig. 3 left, baselines are introduced in Sec. 6). Other methods, such as variational exploration (MAVEN [25]) and individuality emergence (EOI [26]), are slower to learn optimal strategies.

## 5 Related Work

Deep multi-agent reinforcement learning algorithms have witnessed significant advances in recent years. COMA [20], MADDPG [9], PR2 [27], and DOP [10] study the problem of policy-based multi-agent reinforcement learning. They use a (decomposed) centralized critic to calculate gradients for decentralized actors. Value-based algorithms decompose the joint value function into individual utility functions in order to enable efficient optimization and decentralized execution. VDN [4], QMIX [5], and QTRAN [21] progressively expand the representation capabilities of the mixing network. QPLEX [6] implements the full IGM class [21] by encoding the IGM principle into a duplex dueling network architecture. Weighted QMIX [23] proposes weighted projection to decompose any joint action-value functions. There are other works that investigate into MARL from the perspective of coordination graphs [28–30], communication [31, 32, 15], and role-based learning [17, 33].

**Knowledge sharing in MARL** From IQL [34] to QPLEX, many works focus on designing mixing network structures and have provided promising empirical and theoretical results. For these works, experience sharing among agents has been an important component. Learning from others is one essential skill engraved in humans' genes to survive in society. Based on the relationship between teachers and students in human society, a series of research work hopes each agent can learn from others or selectively share its knowledge with others [35–37]. But it is challenging to specify

knowledge in practice, let alone deciding what to share or learn. SEAC [38] partially solves this problem by sharing trajectories only for off-policy training. NCC [32] maintains cognition consistency by representation alignment between neighbors. Roy et al. [39] force each agent to predict others' local policies and adds a coach for group experience alignment. Christianos et al. [40] group agents during pre-training and force agents in the same group to use one policy. In this paper, we do not try to let agents choose whether to learn or share experiences. Our neural network structure shown in Fig. 2 can balance group coordination and diversity by gradient backpropagation.

**Diversity** In single-agent settings, diversity emerges for exploration or solving sparse reward problems. Existing methods such as curiosity-driven algorithms [41–44] or maximising mutual information [45–47] have shown great promise. When encouraging diversity in MARL settings, agents' coordination must be considered. Several recent works study this problem, such as MAVEN [25], EITI & EDTI [48], and EOI [26]. MAVEN learns a diverse ensemble of monotonic approximations with the help of a latent space to explore. EITI and EDTI consider pairwise mutual influence to encourage the interdependence between agents. EOI combines the gradient from the intrinsic value function (IVF) and the total Q-function to train each agent's local Q-function. In this paper, we encourage agents to explore unique trajectories by optimizing the mutual information between agent's identity and trajectory. Moreover, we propose a novel network structure to enable experience sharing or consensus, which combines all agents' rare ideas, while still maintain independent action-value functions for each agent to behave differently when necessary. Our approach considers the trade-off relationship between knowledge sharing and diversity, and learns to establish a balance and leverage their advantages for joint task solving.

## 6 Experiments

In Sec. 4, we use a toy game to illustrate how our approach adaptively balances experience sharing and identity-aware diversity. In this section, we use challenging tasks from GRF and SMAC benchmark to further demonstrate and illustrate the outperformance of our approach. We compare our approach against multi-agent value-based methods (QMIX [5], QPLEX [6]), variational exploration (MAVEN [25]), and individuality emergence (EOI [26]) methods. Different from baselines, we do not include agents' identification in inputs when calculating local Q-functions. We show the average and variance of the performance for our method, baselines, and ablations tested with five random seeds.

### 6.1 Performance on Google Research Football (GRF)

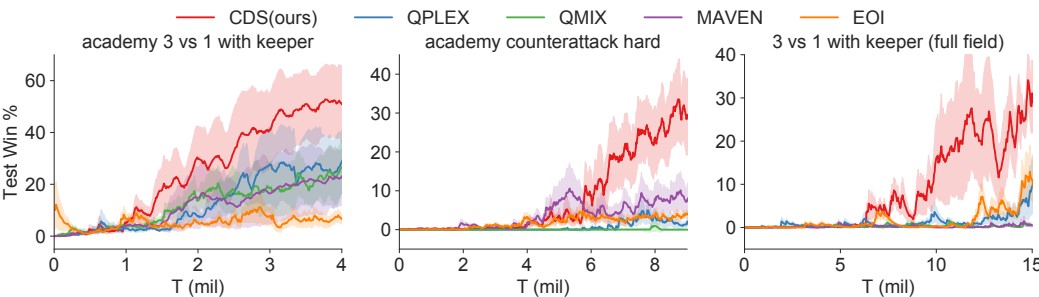

Figure 4: Comparison of our approach against baseline algorithms on Google Research Football.

We first benchmark our approach on three challenging Google Research Football (GRF) offensive scenarios `academy_3_vs_1_with_keeper`, `academy_counterattack_hard`, and our own designed full-field scenario `3_vs_1_with_keeper` (`full field`). Agents' initial locations for each scenario are shown in Appendix B.3. In GRF tasks, agents need to coordinate timing and positions for organizing offense to seize fleeting opportunities, and only scoring leads to rewards. In our experiments, we control left-side players (in yellow) except the goalkeeper. The right-side players are rule-based bots controlled by the game engine. Agents have a discrete action space of 19, including moving in eight directions, sliding, shooting, and passing. The observation contains the positions and moving directions of the ego-agent, other agents, and the ball. The $z$-coordinate of the ball is also included.

We make a small and reasonable change to the half-court offensive scenarios: our players will lose if they or the ball returns to our half-court. All baselines and ablations are tested with this modification. Environmental reward only occurs at the end of the game. They will get +100 if they win, else get -1.

We show the performance comparison against baselines in Fig. 4. Our approach outperforms all the scenarios. MAVEN needs more time to explore sophisticated strategies, demonstrating that CDS incentives more efficient exploration. EOI lets each agent consider individuality and cooperation simultaneously by setting local learning objectives but without exclusive Q networks, making cooperation and individuality hard to be persistently coordinated. In comparison, taking advantage of the partially shared network structure, CDS agents learn diverse but coordinated strategies. For example, as shown in Fig. 1, three agents have different behaviors, with the first agent passing the ball, the second scoring, while the third running to threaten. These diverse behaviors closely coordinate, forming a perfect scoring strategy and leading to significant outperformance against EOI.

## 6.2 Performance on StarCraft II

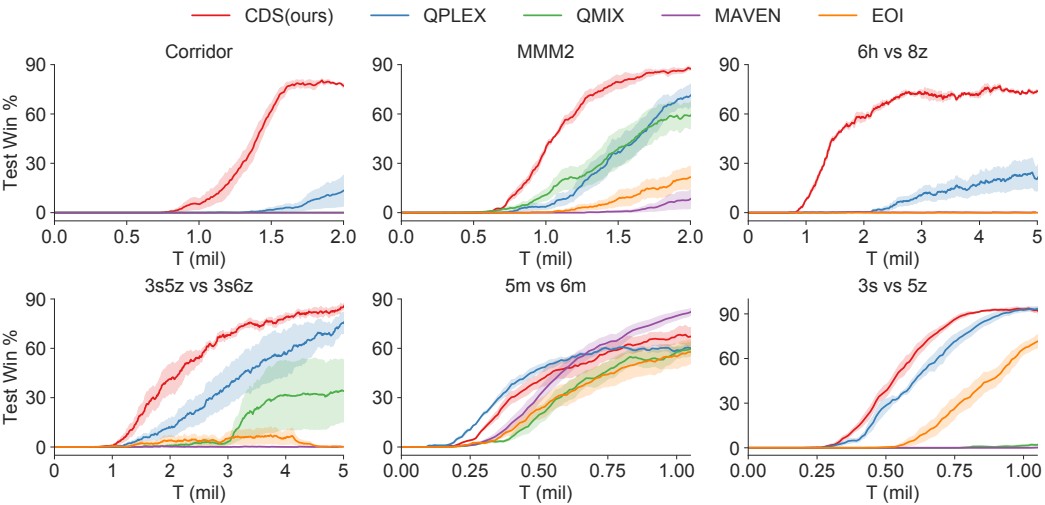

Figure 5: Comparison of our approach against baseline algorithms on four **super hard** SMAC maps: `corridor`, `MMM2`, `6h_vs_8z`, and `3s5z_vs_3s6z` and two **hard** SMAC maps: `5m_vs_6m` and `3s_vs_5z`.

In this section, we test our approach on the StarCraft II micromanagement (SMAC) benchmarks [16]. This benchmark consists of various maps classified as easy, hard, and super hard. Here we test our method on four super hard maps: `corridor`, `MMM2`, `6h_vs_8z`, and `3s5z_vs_3s6z`, and two hard SMAC maps: `5m_vs_6m` and `3s_vs_5z`. For the four super hard maps, our approach outperforms all baselines with acceptable variance across random seeds, as shown in Fig. 5. The baselines QPLEX and QMIX can achieve satisfactory performance on some challenging benchmarks, such as `3s5z_vs_3s6z` and `MMM2`. But on other maps, they need the proposed diversity-celebrating method to get better performance. Compared with MAVEN and EOI, our approach maintains its out-performance with the balance between diversity and homogeneity for learning sophisticated cooperation. Our approach performs similarly with baselines for the two hard maps, indicating our balancing process may not improve the learning efficiency in environments that require pure homogeneity. But for challenging environments, where sophisticated strategies are laborious to explore, our approach can efficiently search for valuable strategies with stable updates.

## 6.3 Ablations and Visualization

To understand the contribution of each component in the proposed CDS framework, we carry out ablation studies to test the contribution of its three main components: Identity-aware *diversity* (A) encouragement and *partially shared* (B) neural network structure with *L1 regularization* (C) on non-shared Q-functions. To test component A, we ablate our intrinsic rewards to four different levels. (1) CDS-Raw ablates all intrinsic rewards by setting $\beta$ in Eq. 8 to zero. (2) CDS-No-Identity ablates

$H\left(\tau_T|id\right)$ and only optimize $H\left(\tau_T\right)$ in Eq. 1 by setting $\beta_1$ in Eq. 6 to zero. (3) CDS-No-Action ablates item ② in Eq. 2 by setting $\beta_2$ in Eq. 6 to zero. (4) CDS-No-Obs ablates item ③ in Eq. 2 by ablating $\beta_1 \log q_\phi\left(o_{t+1}|\tau_t, a_t, id\right) - \log p\left(o_{t+1}|\tau_t, a_t\right)$ in Eq. 6. To test component B, we design CDS-All-Shared, which ablates independent action-value functions together with the L1 loss and, like baselines, adds agents' identification to the input. To test component C, we design CDS-No-L1, which ablates L1 regularization terms by setting $\lambda$ in Eq. 9 to zero.

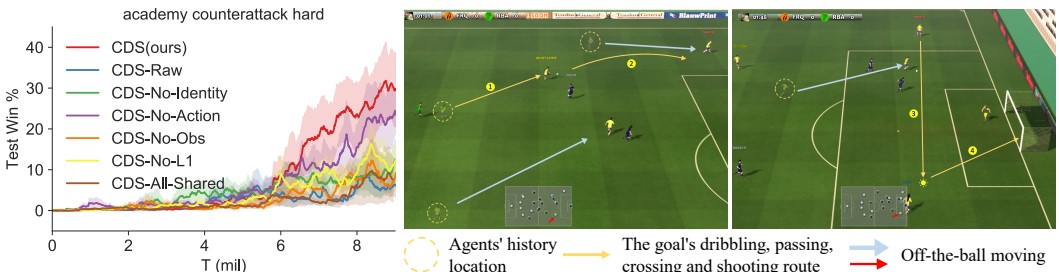

Figure 6: **Left.** Ablation studies on `academy_counterattack_hard`. **Right.** Visualization of trained policies, which achieve complex cooperation with impressive off-the-ball moving strategies.

We first carry out ablation studies on `academy_counterattack_hard` to analyze which part of our novelties lead to the outstanding performance as shown on the left side of Fig. 6. The ablation of each part of our intrinsic reward will bring a noticeable decrease in performance. Among them, the least impact on performance is the ablation of action-aware diversity. CDS-No-L1 performs similarly to MAVEN, which indicates that unlimited diversity is harmful to cooperation. CDS-All-Shared performs even worse than QPLEX, demonstrating that identity-aware diversity is difficult to emerge without our specially designed network structure.

We further visualize the final trained strategies on the right side of Fig. 6, which shows complex cooperation between agents. Our players first attack down the wing by dribbling and passing the ball. Then one of them draws the attention of the enemy defenders and the goalkeeper, while the ball being passed across the penalty area. Another player catches the ball and completes the shot. The most impressive part of our sophisticated strategies is off-the-ball moving strategies. All agents without the ball try to use their unique and valuable moves to create more scoring opportunities, which shows behavior and position diversity for finishing the goal.

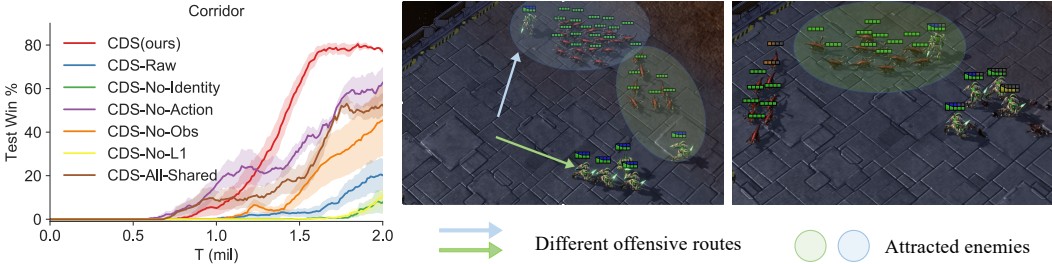

Figure 7: **Left.** Ablation studies in super hard map `corridor`. **Right.** Visualization of the final trained strategies, which achieves a hard-earned victory brought by the sacrifice of a warrior.

We also carry out ablation studies on the super hard map `corridor` as shown in Fig. 7 left. Same as results on `academy_counterattack_hard`, the ablation of action-aware diversity causes the least performance gap. Among all the ablations, CDS-No-L1 and CDS-No-Identity perform worst, whose performance is similar to QPLEX. This phenomenon indicates excessive diversity is harmful to the emerge of complex cooperation. CDS-All-Shared achieves acceptable performance, different from the GRF scenario, reflecting the different demand levels for the representation diversity of these two kinds of benchmarks.

To better explain why our approach performs well. On `corridor`, we also visualize the final strategies in Fig. 7 right. In this super hard map, six friendly Zealots are facing 24 enemy Zerglings. The disparity in quantity means our agents are doomed to lose if they attack together. One Zealot, whose

route is highlighted blue, becomes a warrior leaving the team to attract the attention of most enemies in the blue oval. Although doomed to sacrifice, he brings enough time for the team to eliminate a small part of the enemies in the green oval. After that, another Zealot stands out to attract some enemies and enables teammates to eradicate them. These sophisticated strategies reflect the leverage between diversity and homogeneity by encouraging agents to be diverse only when necessary.

## 7 Closing Remarks

Observing that behavioral diversity among agents is essential for many challenging and complex multi-agent tasks, in this paper, we introduce a novel mechanism of *being diverse when necessary* into shared multi-agent reinforcement learning. The balance between individual diversity and group coordination induced by our CDS approach pushes forward state-of-the-art of deep MARL on challenging benchmark tasks while keeping parameter sharing benefits. We hope that our method can shed light on future works to motivate agents to cooperate with diversity to further explore complex multi-agent coordination problems.

## Acknowledgments

This work was funded by the National Key Research and Development Project of China under Grant 2017YFC0704100 and 2016YFB0901900, in part by the National Natural Science Foundation of China under Grant 61425027 and U1813216, in part by Science and Technology Innovation 2030 – "New Generation Artificial Intelligence" Major Project (No. 2018AAA0100904), a grant from the Institute of Guo Qiang,Tsinghua University, and a grant from Turing AI Institute of Nanjing.

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
