# A  Estimating Intrinsic Reward Functions

In our approach, the intrinsic reward can be separated into two parts. One is related to action-aware diversity, while the other is related to observation-aware diversity. We revisit the formulation of our information-theoretic objective (Eq. 2) and discuss how to better estimate it.

## A.1  Intrinsic Rewards for Action-Aware Diversity

First we analyze term ②, which is related to action-aware diversity. This part can be written as:

$$② = \sum_{t=0}^{T-1} E_{id,\tau} \left[ \log \frac{p(a_t|\tau_t, id)}{p(a_t|\tau_t)} \right] \tag{10}$$

The computational problem here is how to estimate $p\left(a_t|\tau_t\right)$, which can be expanded as:

$$p\left(a_t|\tau_t\right) = \sum_{id} p\left(id|\tau_t\right) \pi\left(a_t|\tau_t, id\right), \tag{11}$$

where $p\left(id|\tau_t\right)$ needs estimation. Any approximation will result in an upper bound of the objective:

$$② = \sum_{t=0}^{T-1} \left( E_{id,\tau} \left[ \log \frac{p(a_t|\tau_t, id)}{q(a_t|\tau_t)} \right] - D_{\text{KL}}\left(p\left(a_t|\tau_t\right) \| q\left(a_t|\tau_t\right)\right) \right) \leq \sum_{t=0}^{T-1} E_{id,\tau} \left[ \log \frac{p(a_t|\tau_t, id)}{q(a_t|\tau_t)} \right]. \tag{12}$$

However, to optimize term ②, we need a lower bound (Eq. 3). Additional approximation of $p\left(id|\tau_t\right)$ may render the optimization intractable. Fortunately, as we will show in this section, a good but not accurate approximation of $p\left(id|\tau_t\right)$ can still lead to satisfactory learning performance. We now discuss two ways to estimate $p\left(id|\tau_t\right)$.

First, we can follow previous work [46] and make the assumption that $p\left(id|\tau_t\right) \approx p\left(id\right)$, which conforms to a uniform distribution. Then, we can calculate $p\left(a_t|\tau_t\right)$ as below:

$$p\left(a_t|\tau_t\right) \approx \frac{1}{n} \sum_{id} \pi\left(a_t|\tau_t, id\right), \tag{13}$$

where $n$ is the number of agents.

However, in this paper, we encourage each agent to behave differently from others when necessary. As a result, it is likely that $p\left(id\right)$ might occasionally be different from $p\left(id|\tau_t\right)$, which means the above assumption might not be valid all the time.

As an alternative, we can leave out the assumption by using the Monte Carlo method (MC) for estimating the distribution $p\left(id|\tau_t\right)$. In tabular cases, we count from samples to calculate the frequency $p\left(id|\tau_t\right) = \frac{N(id,\tau_t)}{N(\tau_t)}$, where $N(\cdot)$ is the time of visitation. However, MC becomes impractical in complex environments such as GRF and SMAC with long horizons and continuous spaces. For these complex cases, inspired

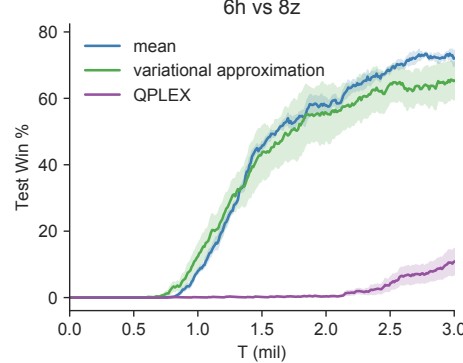

Figure 8: Comparison of assuming $p\left(id|\tau_t\right) \approx p\left(id\right)$ and estimating $p\left(id|\tau_t\right)$ with variational inference on a SMAC super hard map 6h_vs_8z.

by Wang et al. [43], we adopt variational inference to learn a distribution $q_\xi\left(id|\tau_t\right)$, parameterized by a neural network with parameters $\xi$, to estimate $p\left(id|\tau_t\right)$ by optimizing the evidence lower bound (ELBO).

We empirically test these two estimation methods on a SMAC super hard map 6h_vs_8z (Fig. 8) (the observation-aware diversity is estimated with Eq. 15). The experimental results demonstrate that the two methods have similar performance but assuming $p\left(id|\tau_t\right) \approx p\left(id\right)$ outperforms estimating

$p\left(id|\tau_t\right)$ with variational approximation in both average performance and variance between random seeds. We hypothesize that the estimation error renders the variational inference approach unstable. Therefore, we decide to use Eq. 13 to estimate $p\left(a_t|\tau_t\right)$ in this paper.

## A.2 Intrinsic Rewards for Observation-Aware Diversity

We then analyze term ③, which is related to observation-aware diversity. This part can be written as:

$$③ = \sum_{t=0}^{T-1} E_{id,\tau}\left[\log \frac{p(o_{t+1}|\tau_t, a_t, id)}{p(o_{t+1}|\tau_t, a_t)}\right]. \tag{14}$$

The computational problem here is how to estimate $p\left(o_{t+1}|\tau_t, a_t\right)$. We can directly adopt variational inference and learn the variational distribution $q_{\phi_2}\left(o_{t+1}|\tau_t, a_t\right)$, parameterized by a neural network with parameters $\phi_2$, by optimizing the evidence lower bound. With $q_{\phi_2}\left(o_{t+1}|\tau_t, a_t\right)$, $r^I$ can be written as:

$$\begin{aligned}
r^I = E_{id}\,[&\beta_2 D_{\mathrm{KL}}(\mathrm{SoftMax}(\beta_1 Q(\cdot|\tau_t, id))\|p(\cdot|\tau_t)) \\
&+\beta_1 \log q_\phi\left(o_{t+1}|\tau_t, a_t, id\right) - \log q_{\phi_2}\left(o_{t+1}|\tau_t, a_t\right)]. 
\end{aligned} \tag{15}$$

This method use a **forward** prediction model of agents' next observation.

One concern is that the forward method involves inference on the continuous observation space, which may be too large to estimate accurately on complex tasks. We can omit it by deriving a lower bound.

We have that

$$E[\log p(o'|\tau, a, id) - \log p(o'|\tau, a)] = I(o'; id|\tau, a). \tag{16}$$

Therefore, optimising term ③ is equivalent to optimising $I(o'; id|\tau, a)$. Notice that

$$I(o'; id|\tau, a) = H(o'|\tau, a) - H(o'|\tau, a, id), \tag{17}$$

we have

$$\begin{aligned}
I(o'; id|\tau, a) &= -H\left(o'|\tau, a, id\right) + H\left(o'|\tau, a\right) \\
&= -H\left(o'|\tau, a, id\right) + \sum_{o',\tau,a} p(o', \tau, a) \log \frac{p(\tau, a)}{p(o', \tau, a)} \\
&= -H\left(o'|\tau, a, id\right) + \sum_{o',\tau,a} p(o', \tau, a) \log \sum_{id} q\left(id|o', \tau, a\right) \frac{p(\tau, a)}{p(o', \tau, a)} \\
&= -H\left(o'|\tau, a, id\right) + \sum_{o',\tau,a} p(o', \tau, a) \log \sum_{id} q\left(id|o', \tau, a\right) \frac{p(id|o', \tau, a)p(\tau, a)}{p(o', id, \tau, a)} \\
&\geq -H\left(o'|\tau, a, id\right) + \sum_{o',\tau,a} p(o', \tau, a) \sum_{id} p\left(id|o', \tau, a\right) \log \frac{q(id|o', \tau, a)p(\tau, a)}{p(o', id, \tau, a)} \\
&= -H\left(o'|\tau, a, id\right) + \sum_{o',id,\tau,a} p(o', id, \tau, a) \log \frac{q(id|o', \tau, a)}{p(o', id|\tau, a)} \\
&= -H\left(o'|\tau, a, id\right) + E\left[\log q\left(id|o', \tau, a\right)\right] + H\left(o', id|\tau, a\right),
\end{aligned} \tag{18}$$

where $q(id|o', \tau, a)$ can be an arbitrary distribution. We use variational inference and a neural network parameterized by $\eta_1$ to estimate it. Moreover, $H\left(o', id|\tau, a\right)$ can be decomposed as:

$$H\left(o', id|\tau, a\right) = H(id|\tau, a) + H\left(o'|\tau, a, id\right). \tag{19}$$

With Eq. 19, Eq 18 can be further written as:

$$\begin{aligned}
I(o'; id|\tau, a) &\geq -H\left(o'|\tau, a, id\right) + E\left[\log q_{\eta_1}\left(id|o', \tau, a\right)\right] + H\left(o', id|\tau, a\right) \\
&= -H\left(o'|\tau, a, id\right) + E\left[\log q_{\eta_1}\left(id|o', \tau, a\right)\right] + H(id|\tau, a) + H\left(o'|\tau, a, id\right) \\
&= E\left[\log q_{\eta_1}\left(id|o', \tau, a\right)\right] + H(id|\tau, a) \\
&= E\left[\log q_{\eta_1}\left(id|o', \tau, a\right) - \log p(id|\tau, a)\right].
\end{aligned} \tag{20}$$

With the above mathematical derivation, we bypass the estimation of $p\left(o_{t+1}|\tau_t, a_t\right)$. Although $p(id|\tau_t, a_t)$ is introduced, we now infer in a much smaller and discrete space rather than the continuous observation space. We can estimate $p(id|\tau_t, a_t)$ using similar methods introduced in the previous section. We adopt variational inference to learn the distribution $q_{\eta_2}\left(id|\tau_t, a_t\right)$, parameterized by a neural network with parameters $\eta_2$, by optimizing the evidence lower bound. With this **backward** prediction model of agents' identity, $r^I$ can be written as:

$$
\begin{aligned}
r^I = E_{id}\left[\beta_2 D_{\mathrm{KL}}(\mathrm{SoftMax}(\beta_1 Q(\cdot|\tau_t, id))||p(\cdot|\tau_t))\right. \\
\left. +\beta_1 \log q_{\eta_1}\left(id|o_{t+1}, \tau_t, a_t\right) - \log q_{\eta_2}\left(id|\tau_t, a_t\right)\right].
\end{aligned}
\tag{21}
$$

We empirically compare these two methods on a SMAC super hard map `6h_vs_8z` as shown in Fig. 9 ($p(\cdot|\tau_t)$ is estimated with Eq. 13). The experimental results demonstrate that estimating $r^I$ with a forward prediction model noticeably outperforms estimating with a backward prediction model, although the forward model might be more difficult to estimate as discussed before. We hypothesize that simultaneously and independently estimating $q_{\phi}\left(o_{t+1}|\tau_t, a_t, id\right)$ and $q_{\phi_2}\left(o_{t+1}|\tau_t, a_t\right)$ might bring advantages similar to those of curiosity-driven methods, leading to outstanding performance on tasks requiring extensive exploration, so we use Eq. 15 to encourage diversity in this paper.

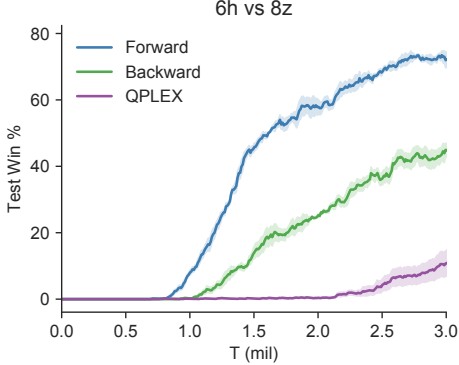

Figure 9: Comparison of forward and backward estimation for observation-aware diversity on a SMAC super hard map `6h_vs_8z`.

## B   Experiment Details

### B.1   Baselines

We compare our approach with multi-agent value-based methods (QMIX [5] & QPLEX [6]), a variational exploration method (MAVEN [25]), and an individuality emergence (EOI [26]) method. For QMIX, QPLEX, MAVEN, and EOI, we use the codes provided by the authors, whose hyper-parameters have been fine-tuned.

### B.2   Architecture and Hyperparameters

In this paper, we use a QPLEX style mixing network with its default hyperparameters suggested by the original paper. Specifically, in complex environments (GRF and SMAC), the transformation part has four 32-bits attentional heads, each with one middle layer of 64 units. Weights in the joint advantage function of the dueling mixing part are produced by a four-head attention module without hidden layers. In toy environment `Pac-Men`, we do not use the transformation part but still generate weights in the joint advantage function of the dueling mixing part by the four-head module without hidden layers. For individual Q-functions, agents share a trajectory encoding network consisting of two layers, a fully connected layer followed by a GRU layer with a 64-dimensional hidden state. After the trajectory encoding network, all agents share a one-layer Q network, while each agent has its independent Q network with the same structure as the shared Q network.

For all experiments, the optimization is conducted using RMSprop with a learning rate of $5 \times 10^{-4}$, $\alpha$ of 0.99, and with no momentum or weight decay. For exploration, we use $\epsilon$-greedy, with $\epsilon$ annealed linearly from 1.0 to 0.05 over $500K$ time steps and kept constant for the rest of the training, for both CDS and all the baselines and ablations. Our method introduces four important hyperparameters: $\beta$, $\beta_1$, $\beta_2$, that are related to the intrinsic rewards, and $\lambda$, which is the scaling weight of the L1 regularization term. For the environment in the case study (Pac-Men), $\lambda$ is set to 0.01. For GRF and SMAC, we set $\lambda$ to 0.1 to strengthen *being diverse only when necessary*. For other hyperparameters related to intrinsic rewards, we tune with grid search ($\beta \in \{0.05, 0.1, 0.2\}, \beta_1 \in \{0.5, 1.0, 2.0\}$, and $\beta_2 \in \{0.5, 1.0, 2.0\}$). We tune for GRF on the `academy_3_vs_1_with_keeper`. We notice `Corridor` has a fundamental difference compared with `6h_vs_8z`, MMM2 and `3s5z_vs_3s6z` in the ratio of the number of enemy agents to our agents, which

requires different kinds of sophisticated strategies. Our approach can be compatible with different situations by adjusting the ratio of $H(\tau_T)$ and $H(\tau_T \mid id)$ in Eq. 1. So we tune hyperparameters for SMAC on both 6h_vs_8z and Corridor. For other SMAC maps, we fine-tune $\beta$ to get better adaptability to the environments. Moreover, for GRF scenarios, we use a prioritized replay buffer of the TD error for CDS, all the baselines, and ablations.

Table 1: CDS Hyperparameters.

|  | Environment | $\beta$ | $\beta_1$ | $\beta_2$ | $\lambda$ |
|---|---|---|---|---|---|
| Case Study | Pac-Men | 0.15 | 2.0 | 1.0 | 0.01 |
| SMAC | 6h_vs_8z | 0.1 | 2.0 | 1.0 | 0.1 |
|  | MMM2 | 0.07 | 2.0 | 1.0 | 0.1 |
|  | 3s5z_vs_3s6z | 0.03 | 2.0 | 1.0 | 0.1 |
|  | Corridor | 0.1 | 0.5 | 0.5 | 0.1 |
|  | 3s_vs_5z | 0.04 | 0.5 | 0.5 | 0.1 |
|  | 5m_vs_6m | 0.01 | 2.0 | 1.0 | 0.1 |
| GRF | academy_3_vs_1_with_keeper | 0.1 | 0.5 | 1.0 | 0.1 |
|  | academy_counterattack_hard | 0.1 | 0.5 | 1.0 | 0.1 |
|  | 3_vs_1_with_keeper (full field) | 0.1 | 0.5 | 1.0 | 0.1 |

## B.3  GRF Scenarios

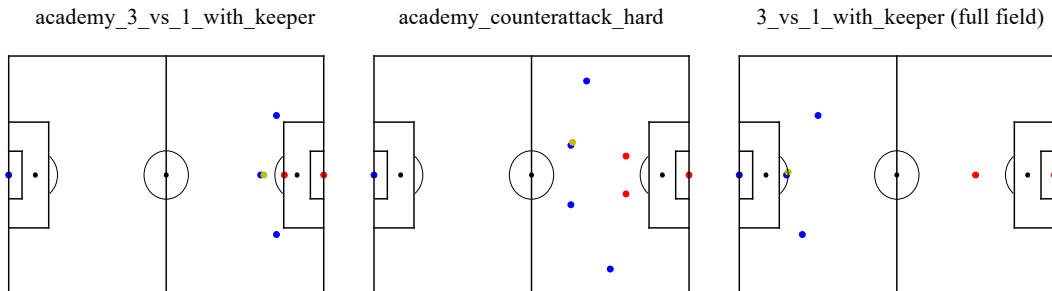

Figure 10: Visualization of the initial position of each agent in three GRF environments considered in our paper, where blue points represent our agents, red points represent opponents, and the yellow point represents the ball.

In this paper, we achieve state-of-the-art performance on all the tested GRF tasks, including two official scenarios academy_3_vs_1_with_keeper and academy_counterattack_hard. Furthermore, we design one full-field scenario 3_vs_1_with_keeper (full field) to compare the performance of our approach and baselines on a task with a more complex problem space. The visualization of the initial position of each agent for the three GRF scenarios are shown in Fig. 10.

## B.4  Infrastructure

Experiments are carried out on NVIDIA GTX 2080 Ti GPU. And the training of our approach on all environments can be finished in less than two days.