# OpenReview forum: "Celebrating Diversity in Shared Multi-Agent Reinforcement Learning"
_NeurIPS.cc/2021/Conference — NeurIPS 2021 Poster_

### Official Review · Reviewer_Rpsf · 2021-07-13

**Rating:** 7
**Confidence:** 3

**Summary:**

The paper deals with the problem of imposing policy diversity along the different agents of a MARL problem. To do so, the paper proposes to introduce a novel objective to maximize the mutual information between the agents identities and trajectories. Additionally, the paper proposes a decoupled Q function architecture to represent shared and non-shared goals.

**Limitations And Societal Impact:**

Not applicable.

**Main Review:**

The paper is well written and presents both quantitative and qualitative metrics to show the performance of the proposed method. Additionally, the paper introduces an insightful ablation study to evaluate the effect of the proposed elements in the learning performance.

However, I think there are few points that should be clarified:

- In Line 120, the paper claims the intractability of the Term 3, given an integral term, but the Term 3 is not explicitly showing any integral term in Equation 2.

- Equation 5, presents the gradient over a Loss function, but this loss function has not been previously introduced properly.

- Even if the paper proposes a decoupled architecture between shared and non-shared Q functions, the learning is done jointly. The paper claims that L1 regularization over the independent Q function, allows the learning to be balanced between the shared and non-shared terms. Can the authors provide any theoretical guarantee of that claim or only experimental?

**Time Spent Reviewing:**

5

---

> ### Author Response · Authors · 2021-08-10
> **Response to Reviewer Rpsf**
>
> We thank the reviewer for the insightful review.
>
> **Q1:** In Line 120, the paper claims the intractability of the Term 3, given an integral term, but the Term 3 is not explicitly showing any integral term in Equation 2.
>
> **A1:** The integral term is introduced when we expand Term 3 as: $p\left(o_{t+1} \mid \tau_{t}, a_{t}, i d\right) = \int p(o_{t+1} \mid s_{t+1}, id)p(s_{t+1} \mid \tau_{t}, a_{t}, i d) ds_{t+1}$. The unknown underlying dynamics and the integral over a continuous space lead to intractable computation, so we use variational inference for estimation. As suggested by the reviewer, we will clarify how the integral is involved in the next version.
>
>
> **Q2:** Equation 5, presents the gradient over a Loss function, but this loss function has not been previously introduced properly.
>
> **A2:** The loss function is $D_{\mathrm{KL}}\left(p(\cdot \mid \tau, a, i d) \| \| q_{\phi}(\cdot \mid \tau, a, i d)\right)$. We optimize this for tightening the estimation gap introduced in Eq. 4. As suggested by the reviewer, we will refine the presentation of the loss function in Eq. 5 in the next version.
>
> **Q3:** Even if the paper proposes a decoupled architecture between shared and non-shared Q functions, the learning is done jointly. The paper claims that L1 regularization over the independent Q function, allows the learning to be balanced between the shared and non-shared terms. Can the authors provide any theoretical guarantee of that claim or only experimental?
>
> **A3:** In this paper, we illustrate this balance based on experiments (Fig. 3). Our ablation studies also demonstrate that the L1 regularization term is essential for learning sophisticated strategies (Fig. 7 and 8). As for the theoretical guarantee, we will explore it in the future.

---

### Official Review · Reviewer_yUaF · 2021-07-14

**Rating:** 7
**Confidence:** 4

**Summary:**

This paper focuses on the centralized training with decentralized execution (CTDE) setting within cooperative multi agent reinforcement learning (MARL). The authors observe that in some cases training with a joint Q function can lead to myopic agents that do not effectively coordinate with one another. The authors introduce CDS, a new method that factorizes the Q function into shared and individual components, with an additional loss term maximizing the mutual information between the individual Q-function and the agent ID. They demonstrate the effectiveness of the approach on a toy problem and then outperform a set of strong baselines on two challenging benchmarks. Overall the work seems like a solid contribution to an active area of research.

I vote to weak accept the paper, with a possible increase if my questions below are answered in a satisfactory manner. The only reason why this paper is not an accept is a general impression that the authors prioritized SoTA over transparency. There is no discussion of how hyperparameters were tuned, very limited discussion of limitations and no discussion of how environments were chosen. Overall it leaves the impression that maybe the method doesn't generalize to other environments, and is brittle to hyperparameters.

**Limitations And Societal Impact:**

Limitations should be discussed more adequately in the main body of the paper. Societal impact is likely minimal.

**Main Review:**

Strengths of the paper:

- It is well-written and motivated. The ideas fit well with the current literature.

- The idea of optimizing mutual information between agent ID and the trajectory is intuitive, and the breakdown into action and observation diversity is well done.

- The toy example is well presented and explained. In particular, the result that the agents follow their local Q-functions along the paths then use the shared one for the rooms is interesting and certainly demonstrates the method effectively.

- The experimental results are strong, outperforming recent (and relevant) baselines, and on both GRF and SMAC (vs. usually just SMAC for related works).

Weaknesses/Questions:

1) How many seeds were used for the pacman task? What does the plot actually show? Mean/median? What are the error bars?

2) The paper introduces four new hyperparameters, all of which were tuned for different environments. There are no ablations showing the sensitivity of the performance to these hyperparameters. How were they chosen? How brittle is the algorithm to these changes?

3) It seems for SMAC the parameters were individually tuned per task, was there not a single parameter configuration that could perform well on all three tasks? This is a separate question from above because it also highlights another issue - how does CDS perform on other tasks? The authors mention that it is not expensive to run their algorithm, yet only use four StarCraft maps, which is fewer than other works. Then for GRF, one of the three environments chosen was hand designed, presumably to favor this specific approach.

4) For GRF, how were the number of timesteps chosen? It looks to be arbitrary, and makes it appear that the other methods may outperform later on.

5) The language is, at times, a little overzealous. In particular, referring to the experimental results as "extraordinary" (l.62). I don't think this is needed, *the results are strong as they are without sensationalizing them*.

6) The limitations of the method are barely mentioned. There is one paragraph at the end of the appendix, and 50% of it is reiterating the strength of the approach. It would be useful to understand when this fails. I think this gives the general impression that the authors are trying to only show strengths, maybe through fear of reviewers criticising weaknesses. In my opinion it would actually be better for the paper to show when it does not work, so that future research can be done to improve upon it.

7)  The Appendix shows the objective is actually using the mean over IDs, which the authors say themselves should not be accurate, so it is possible they tuned hyperparameters to a noisy objective and picked the best result.

Minor issues and typos (did not impact score):

- l.16 "provides a promise" → "provides promise"

- l.63 "achieve" → "achieves"

- l. 64 "talent" → "talented"

- l.105 "between individual" → "between an individual"

- l. 112 "about agent's" → "about an agent's"

- l.335 "into" → "in"

**Time Spent Reviewing:**

5

---

> ### Author Response · Authors · 2021-08-10
> **Response to Reviewer yUaF**
>
> We thank the reviewer for the insightful review.
>
> **Q1:** How many seeds were used for the pacman task? What does the plot actually show? Mean/median? What are the error bars?
>
> **A1:**  We show the average and variance of episode returns for our method and baselines, which are evaluated with five random seeds for the pacman task.
>
> **Q2.1:** How were hyperparameters chosen?
>
> **A2.1:**  Our approach introduces four new hyperparameters, including the weight of the L1 regularization term, which is fixed to 0.1 and not tuned for both GRF and SMAC. For other hyperparameters related to intrinsic rewards, we tune for GRF on the *academy 3 vs 1 with keeper* and tune for SMAC on the *6h vs 8z* and *Corridor* with grid search ($\beta \in$ {0.05, 0.1, 0.2}, $\beta_1 \in$ {0.5, 1.0, 2.0}, and $\beta_2 \in $ {0.5, 1.0, 2.0}).
>
>
> **Q2.2:** How brittle is the algorithm to hyperparameters?
>
> **A2.2:**  We compare the average and best performance during tuning shown below. Different hyperparameters lead to various performances. But our algorithm can significantly improve learning efficiency compared with baselines after a rough selection of hyperparameters. Moreover, the average performance is already better than some baselines.
>
>
> |         Environment        |            |   |     |
> |:--------------------------:|:----------:|:-------------------------:|:---:|
> | academy 3 vs 1 with keeper | time steps |             2M            |  4M |
> |                            |     avg    |             6%            | 15% |
> |                            |    best    |            14%            | 46% |
> |          6h vs 8z          | time steps |             1M            |  2M |
> |                            |     avg    |             3%            | 16% |
> |                            |    best    |             8%            | 61% |
> |          Corridor          | time steps |             1M            |  2M |
> |                            |     avg    |             1%            | 11% |
> |                            |    best    |             5%            | 78% |
>
>
> **Q3.1:** Was there not a single parameter configuration that could perform well on all tasks in SMAC?
>
> **A3.1:** We notice *Corridor* has a fundamental difference compared with *6h vs 8z*, *MMM2* and *3s5z vs 3s6z* in the ratio of the number of enemy agents to our agents, which requires different kinds of sophisticated strategies. Our approach can be compatible with different situations by adjusting the ratio of $H\left(\tau_{T}\right)$ and $H\left(\tau_{T} \mid i d\right)$ (Eq. 1). We show the performance of the four maps used in our paper with {$\beta, \beta_{1}, \beta_{2}$}={0.1,2.0,1.0} as below.
>
> | Corridor | 1M  | 2M  | MMM2 | 1M   | 2M   | 6h vs 8z | 2.5M   | 5M   | 3s5z vs 3s6z | 2.5M | 5M  |
> |----------|-----|-----|------|------|------|------|------|------|--------------|------|-----|
> |          | 0%  | 2%  |      | 32%  | 81%  |      | 70%  | 76%  |              | 27%  | 62% |
>
> **Q3.2:** How does CDS perform on other tasks? The authors mention that it is not expensive to run their algorithm, yet only use four StarCraft maps, which is fewer than other works. Then for GRF, one of the three environments chosen was hand-designed, presumably to favor this specific approach.
>
> **A3.2:** For SMAC, our approach can accelerate the learning of sophisticated strategies in challenging environments but plays a minor role in other environments. In the paper, we have demonstrated the strength of our approach in 4 out of 5 super hard maps. Besides that, we have also run hard maps such as *3s vs 5z* and *5m vs 6m*, where CDS performs similarly with QPLEX with a performance improvement of close to 3%. We hypothesize whether our approach can obviously accelerate learning depends on the difficulty of sophisticated strategies being explored. We will add these experimental results and analysis to the next version of our paper for a more thorough analysis of our approach's applicable scenes.
>
> For GRF, the hand-designed scenario extends the football field from half to full. This scenario shows how performance changes when the action-observation space grows.
>
> **Q4:** For GRF, how were the number of time steps chosen?
>
> **A4:** Based on the time and resources limitations, we set time steps according to the number of agents and the size of the field.
>
>
> **Q5:** The limitations of the method are barely mentioned.
>
> **A5:** As suggested by the reviewer, we will reorganize the paper's content and put limitations and potential negative impact of our approach in the main text to discuss. Here we relist them as below.
>
> - Our approach attempts to achieve a balance between diversity and sharing to enable effective cooperative learning. But for environments that require pure diversity or homogeneity, this balancing process may not improve the learning efficiency.
>
> - As suggested by the reviewer, we establish our approach based on the CTDE framework, which is currently unsuitable when agents need to cooperate with unknown agents or humans. We will explore whether our approach is suitable for other multi-agent frameworks in the future.
>
> - Diversity introduced in our approach might cause unintentional damage or cost when exploring reality, which might need more safety constraints.
>
> - As for the potential negative social impact, we are worried about the possibility of our approach being used by others in the military field, for example, the control of automatic weapons and UAV teams. We hope that multi-agent cooperation is promoted for social goods rather than any violent usage.
>
> **Q6:** The Appendix shows the objective is actually using the mean over IDs, which the authors say themselves should not be accurate, so it is possible they tuned hyperparameters to a noisy objective and picked the best result.
>
> **A6:** It is a common practice to assume $p\left(i d \mid \tau_{t}\right) \approx p(i d)$ [1, 2, 3]. In the appendix, we explore another way for estimating $p\left(i d \mid \tau_{t}\right)$ with neural networks, which is noisier (the estimation error during training is never close to 0, with a minimum value of 1.34, in the *6h vs 8z* environment). So we choose the mean over IDs actually for decreasing the estimation noise.
>
> [1] Sharma, A., Gu, S., Levine, S., Kumar, V., & Hausman, K. (2019, September). Dynamics-Aware Unsupervised Discovery of Skills. In International Conference on Learning Representations.
>
> [2] Kim, J., Park, S., & Kim, G. (2021, July). Unsupervised Skill Discovery with Bottleneck Option Learning. In International Conference on Machine Learning (pp. 5572-5582). PMLR.
>
> [3] Osa, T., Tangkaratt, V., & Sugiyama, M. (2021). Discovering Diverse Solutions in Deep Reinforcement Learning. arXiv preprint arXiv:2103.07084.
>
>
> **Minor issues and typos**
>
> We appreciate your comments and will incorporate them in the next version of our paper.

---

> > ### Comment · Reviewer_yUaF · 2021-08-15
> > **Raising my score**
> >
> > Thank you for providing this information.
> >
> > It would be great if the details around hyperparameters could be included in the Appendix, with more details on the experiments and limitations in the main body. Now that it is clearer, I am happy to raise to an accept.
> >
> > This paper should be an interesting contribution to the multi-agent community as well as those interested in diversity in RL more generally.

---

> > > ### Author Response · Authors · 2021-08-15
> > > **Thanks for the response**
> > >
> > > Thanks for your valuable comments and suggestions. We will incorporate them into our next revision.

---

### Official Review · Reviewer_zJ1w · 2021-07-15

**Rating:** 6
**Confidence:** 4

**Summary:**

This paper proposes a CDS method that adaptively trades off diversity and sharing in multi-agent reinforcement learning. The authors introduce an information-theoretical objective to encourage diversity and decompose individual Q-functions as the sum of shared and non-shared local Q-functions to boost shared knowledge usage whenever possible. Visualizations show that the balance between identity-aware diversity and homogeneity promotes sophisticated strategies.

**Ethical Concerns:**

There are no ethical concerns with this paper.

**Ethics Review Area:**

["I don’t know"]

**Limitations And Societal Impact:**

The authors do not address the limitations of their work, but I have some questions and suggestions about this paper:

1. The essence of exploration should be for the team as a whole to interact thoroughly with the environment, so why should the agents within a team have to behave differently? How does the algorithm perform in scenarios that require multiple agents to work together on a single task (for instance, a door that requires two agents to open together)?

2. Figure 3 is attractive and illustrates some of the intrinsic mechanics and advancements of the method, but I did not understand the role of independent Q very well with this example. Is the passage of agents through different paths a reflection of different strategies? Passing diverse paths can also be summarized macroscopically as the same strategy. I suggest designing more complex scenarios, such as some agents passing through paths while others need to stay in the central room to accomplish the task together. These may be more like manifestations of different strategies.

3. Why were the agents' identifications not included when calculating the local Q-functions in the experiments?

4. Figure 5 shows why the CDS method works, illustrating the process of training in which the agents first personalize, then assimilate, and finally find a balance between diversity and homogeneity. This visualization, although interesting, does not seem to be a convincing illustration of this method. Because this process can also be explained by the fact that the agents first focus on self-exploration and then learn to cooperate, which is the same with the vast majority of MARL approaches. An ablation experiment is a more proper way to prove this. I suggest removing modules that encourage individualization and assimilation, respectively, and visualize how these cases perform at the same number of training steps. These experiments can show that the CDS is effective because of its balance between diversity and homogeneity.

**Main Review:**

Originality: It is novel and interesting to introduce diversity into the common-used parameter sharing technique. The method is new and differs from previous works in this area. The related works are adequately cited and reasonably compared.

Quality: I have not checked all details, but the paper appears to be technically sound. The experimental results illustrate the method’s superiority to some extent, but some visualizations may not support some of the ideas in the paper very well.

Clarity: This paper is clearly written and well organized. It provides the reader with some essential information to reproduce its results.

Significance: This submission introduces diversity into the parameter sharing technique, and to some extent, advances the field of multi-agent systems. Other researchers are likely to use the ideas in this paper.

**Time Spent Reviewing:**

6

---

> ### Author Response · Authors · 2021-08-10
> **Response to Reviewer zJ1w**
>
> We thank the reviewer for the thoughtful comments.
>
> **Q1.1:**  The essence of exploration should be for the team as a whole to interact thoroughly with the environment, so why should the agents within a team have to behave differently?
>
> **A1.1:** Encouraging agents to behave differently is a possible way to induce more thorough interactions with the environment. Particularly when agents share learning, their behaviors tend to be homogeneous, which limits agents' exploration as a whole team. This is one of the reasons that CDS performs significantly better than MAVEN, which directly encourages exploration for the team as a whole (Fig. 4 and Fig. 6 in our paper).
>
> **Q1.2:** How does the algorithm perform in scenarios that require multiple agents to work together on a single task (for instance, a door that requires two agents to open together)?
>
> **A1.2:** As commented by the reviewer, some multi-agent tasks require homogeneous policies. In the meantime, others need diverse policies or in between. To better solve a wider range of tasks, we propose to encourage diversity only when necessary. In the reviewer's example, CDS agents can use the shared Q function to act similarly. This claim is supported by the observation that CDS agents have the ability to attack the same target in SMAC for victories.
>
> **Q2:**  Figure 3 is attractive and illustrates some of the intrinsic mechanics and advancements of the method, but passing diverse paths can also be summarized macroscopically as the same strategy. I suggest designing more complex scenarios, such as some agents passing through paths while others need to stay in the central room to accomplish the task together.
>
> **A2:** For the scene the reviewer describes, we sincerely think it will be more challenging and require diverse strategies on a more macroscopical scale, which is also interesting for us.  So we create a new environment based on the original *pacmen* task, where we add a button to the central room, which moves randomly every 20 steps. Only when one agent steps on this button can the dots in other rooms be eaten. In addition, we add one more agent and shorten the distance of all paths. The performance comparison between our algorithm and QMIX is shown below.
>
> | Method | 2M | 4M | 6M | 8M |
> |--------|----|----|----|----|
> | CDS    | -6 | -3 | 18 | 94 |
> | QMIX   | -8 | -5 | 3  | 9  |
>
> Our original case study has illustrated that agents driven by our intrinsic rewards can achieve reasonable action-aware and observation-aware diversity while sharing sufficient experience. The new experimental results demonstrate that our approach can encourage agents to achieve diversity on a more macroscopical scale. We will add the visualization of different strategies in this new environment in the next version.
>
> **Q3:**  Why were the agents' identifications not included when calculating the local Q-functions in the experiments?
>
> **A3:**  For standard MARL algorithms with shared parameters, agents' identifications are included when calculating the local Q-functions for representation diversity. Independent Q-functions can achieve a similar goal in our approach, so we abandon this information. As suggested by the reviewer, we add agents' identifications when calculating the local Q-functions and test on the *6h vs 8z* map. Experimental results demonstrate that adding this information will improve the performance by 8% in this environment.
>
> **Q4:**  I suggest removing modules that encourage individualization and assimilation, respectively, and visualize how these cases perform at the same number of training steps. These experiments can show that the CDS is effective because of its balance between diversity and homogeneity.
>
> **A4:**  We have ablated all intrinsic rewards encouraging individualization (CDS-Raw) and L1 regularization terms encouraging assimilation (CDS-No-L1) in Fig. 7 and Fig. 8. Both ablations lead to performance decrease, which shows the importance of our modules encouraging individualization and assimilation.
>
> Furthermore, we add one more ablation (CDS-NoL1-Noshare), which ablates the L1 regularization term and the shared Q-function. In this way, all modules encouraging assimilation are removed. We test this ablation on *Corridor*, which loses all the games in evaluation even after 2M training time steps. This ablation shows the significance of shared learning for encouraging assimilation. We will provide a detailed comparison against this ablation and visualization of our approach in the next version of our paper.

---

> > ### Comment · Reviewer_zJ1w · 2021-08-27
> > **Acknowledgement of response**
> >
> > Thank you very much for the detailed responses. I have no more concerns about the manuscript. Please incorporate these responses selectively into the final version of the manuscript.

---

> > > ### Author Response · Authors · 2021-08-27
> > > **Thanks for the response**
> > >
> > > Thank you very much for your positive and thoughtful comments. We will incorporate your suggestions into our next revision.

---

### Official Review · Reviewer_u9nV · 2021-07-16

**Rating:** 8
**Confidence:** 5

**Summary:**

Incentivizes agents to be diverse from each other in cooperative multi-agent tasks by getting them to maximize the mutual information between their trajectories and their ID. Breaks this down into action-diversity (maximizing the KL divergence between each agent’s action distribution & the group average), as well as observation-diversity (using a variational approximation of I(o_t;ID|tau,a_t)). Also propose a new architecture that allows sharing of information between agents, while retaining some agent-specific parameters. Thorough evaluation with 6 state-of-the-art multi-agent baselines show the method gives consistently much higher performance than prior techniques at both Google Research Football and Starcraft. Great analysis of the learned behaviors and why diversity helps in both environments, cool ablation studies.


**Limitations And Societal Impact:**

The introduction clearly identifies real-world problems for which this technique could provide social benefit, but the paper does not include a discussion of limitations and potential negative impact in the main text. It might be worth mentioning the limitation that because the method relies on centralized training, it is not applicable to multi-agent problems in which some of the other agents are humans, and thus are outside of the control of the learning algorithm. E.g. it is not applicable to autonomous driving.


**Main Review:**

**Quality:**
- The paper proposes a well-motivated and effective technique for improving cooperation and learning in multi-agent RL (MARL). It combines a (somewhat) novel diversity objective with a novel architecture that enables sharing information between agents while allowing for agent-specific policies.
-  The experimental evaluation is extremely thorough, and includes 6 baseline methods spanning state-of-the-art multi-agent techniques. The performance gains over these methods are convincing, across 2 challenging MARL environments.
- The paper also provides experiments in a 3rd environment which provide insight into the method.
- Thorough ablation studies analyze exactly which components of the algorithm contribute to performance. Interestingly, this differs between the multi-agent environments under consideration.

**Significance:**
- Given the magnitude of the performance between this technique and several SOTA baselines, this paper is definitely of interest to the MARL community
- The progress on GRF and the analysis of how diversity contributes to this is also compelling.

**Originality/Novelty:**
- The biggest issue with the paper is novelty. Similar diversity objectives have been proposed in DIAYN (https://arxiv.org/abs/1802.06070), and in the multi-agent case with EOI (https://arxiv.org/pdf/2006.05842.pdf) and CMRL (https://arxiv.org/abs/1903.02710). However, the paper benchmarks against EOI and shows convincing performance improvements above this method. DIAYN is appropriately cited.

**Clarity:**
- The paper is well motivated, and the introduction is clear and compelling.
- Intro figure clearly sets up the issue with learning shared policies and why diversity is important.
- The derivation of the method in section 3.1 is very clear and crisp, especially highlighting each of the three terms in equation 2.
- However, it would improve clarity to briefly describe (even in 1-2 sentences) how p(a_t|\tau_t) and p(o_{t+1}|\tau_t,a_t) are estimated in the main text.
- The L1 regularization could be more clearly motivated. Why L1 not L2?
- Figure 3 would be much more compelling if it included a heatmap for one of the baseline methods (as in the EOI paper). This helps make the case for the importance of explicitly incentivizing diversity, rather than just optimizing shared reward.
- Lines 255-260 talk about how during phase 2 of learning in GRF, the agents "lose their individuality" to focus on scoring. And yet even in this phase, the performance of CDS is significantly higher than the baselines (~25% vs~10%). If the agents have lost their individuality, what explains this performance gap?

**Detailed feedback/suggestions:**
- Lines 54-57 in the intro could be made more clear/crisp.
- Although the results of the paper are convincing, calling your own results "extraordinary" (line 62) might be considered a bit grandiose.
- "agents need to behave differently to /highlight/ themselves from others" -> not proper grammar. "distinguish themselves from others" would be more appropriate
- When referring to QPLEX in line 153 you should include the citation of the QPLEX paper.
- The idea of rewarding agents for mutual influence was first proposed in https://arxiv.org/abs/1810.08647. It would be appropriate to cite this work in lines 208-209 of the related work when introducing EITI and EDTI.

**Time Spent Reviewing:**

3.5

---

> ### Author Response · Authors · 2021-08-10
> **Response to Reviewer u9nV**
>
> We thank the reviewer for the inspiring comments.
>
> **Q1:** The biggest issue with the paper is novelty. Similar diversity objectives have been proposed in DIAYN, and in the multi-agent case with EOI and CMRL. However, the paper benchmarks against EOI and shows convincing performance improvements above this method. DIAYN is appropriately cited.
>
> **A1:** Previous related works have shown the promise of achieving diversity. However, we found that excess diversity might be harmful to MARL in joint tasks (see ablation studies in Fig. 8). Our work considers the trade-off relationship between knowledge sharing and diversity, and learns to establish a balance and leverage their advantages for joint task solving. It takes a step further in improving the compatibility of diversity in multi-agent reinforcement learning.
>
> **Q2:**  The L1 regularization could be more clearly motivated. Why L1 not L2?
>
> **A2:** We expect agents to be diverse only when necessary, i.e., they use the shared Q function most of the time. We achieve this by encouraging individual Q values to be prominent only in few situations with a preference for few actions. Compared to L2, an L1 regularization encourages sparsity and is thus suitable for our aim. We will add related discussions in the next version.
>
> **Q3:**  Lines 255-260 talk about how during phase 2 of learning in GRF, the agents "lose their individuality" to focus on scoring. And yet even in this phase, the performance of CDS is significantly higher than the baselines (25 vs 10). If the agents have lost their individuality, what explains this performance gap?
>
> **A3:** In the first phase, diverse exploration can help agents collect promising trajectories to realize the importance of scoring faster. As a result, agents tend to compete for the ball to score. Many baselines learn such a policy at the end of training. For comparison, in the second phase, CDS still continually encourages diverse exploration, and agents gradually have relatively less homogeneous learning than baselines.
>
> **Q4:** The paper does not include a discussion of limitations and potential negative impact in the main text.
>
> **A4:** As suggested by the reviewer, we will reorganize the paper's content and put limitations and potential negative impact of our approach in the main text to discuss. Here we relist them as below.
>
> - Our approach attempts to achieve a balance between diversity and sharing to enable effective cooperative learning. But for environments that require pure diversity or homogeneity, this balancing process may not improve the learning efficiency.
>
> - As suggested by the reviewer, we establish our approach based on the CTDE framework, which is currently unsuitable when agents need to cooperate with unknown agents or humans. We will explore whether our approach is suitable for other multi-agent frameworks in the future.
>
> - Diversity introduced in our approach might cause unintentional damage or cost when exploring reality, which might need more safety constraints.
>
> - As for the potential negative social impact, we are worried about the possibility of our approach being used by others in the military field, e.g., the control of automatic weapons and UAV teams. We hope that multi-agent cooperation is promoted for social goods rather than any violent usage.
>
> **Detailed feedback/suggestions**
>
> We appreciate and will take all the detailed suggestions in the next version of our paper.

---

> > ### Comment · Reviewer_u9nV · 2021-08-10
> > **Acknowledgement of response**
> >
> > Thank you for your response. Adding the provided explanations to the paper will make it more clear.
> >
> > I appreciate the nuanced discussion that optimizing for diversity can be taken too far.
> >
> > > As for the potential negative social impact, we are worried about the possibility of our approach being used by others in the military field, e.g., the control of automatic weapons and UAV teams. We hope that multi-agent cooperation is promoted for social goods rather than any violent usage.
> >
> > Well said! It would be good to include this discussion.

---

> > > ### Author Response · Authors · 2021-08-15
> > > **Thanks for the response**
> > >
> > > Thanks for the inspirational comments and suggestions. We hope our work can bring more thinking of diversity in the multi-agent setting for social goods.

---

### Decision · Program_Chairs · 2021-09-27

**Decision:**

Accept (Poster)

**Comment:**

The authors present a novel approach to cooperative multi-agent reinforcement learning that focuses on balancing diverse behaviors with shared information. The approach incorporates several new insights (mutual information losses to encourage action and observation diversity, shared and local architecture components and a regularization term) and conceptually and empirically demonstrate how each contributes to the algorithms performance. Empirical validation is extensive and includes challenging environments, often showing dramatic performance improvements compared to baselines. Detailed ablations and analysis provide further insights into how the proposed approach works.

Initial reviews were positive, highlighting the novel insights and strong empirical results achieved in the paper. The paper was assessed as well written and largely clear. Open questions included experimental details (e.g., the procedure for hyper-parameter tuning and impact of hyperparameters on performance), positioning of the paper relative to related work (e.g., DIAYN, and the general question of how this work relates to previous approaches that encourage diversity), and conceptual questions (e.g., potential conflicts between diverse behavior and the need for coordinated exploration, the role of L1 regularization over the independent Q functions).

The authors addressed reviewer questions and suggestions in detail, providing clarifications as well as additional empirical results. As a result, several reviewers increased their scores and all indicated that their concerns and questions had been addressed. The AC agrees with this consensus.